# Erdheim–Chester Disease: Investigating the Correlation between Targeted Treatment Therapy and Disease Outcomes

**DOI:** 10.3390/cancers16071299

**Published:** 2024-03-27

**Authors:** Sabrina R. Wilcox, Samuel B. Reynolds, Asra Z. Ahmed

**Affiliations:** Division of Hematology/Oncology, Department of Internal Medicine, University of Michigan Health System, Ann Arbor, MI 48109-5368, USA; reynosam@med.umich.edu (S.B.R.);

**Keywords:** Erdheim–Chester, histiocyte, monocyte

## Abstract

**Simple Summary:**

Erdheim–Chester disease (ECD) is a rare, non-Langerhans cell histiocytic disease, characterized as a clonal hematopoietic malignancy in 2016. MAP kinase and PI3K-AKT pathway somatic mutations and/or fusion genes play significant roles in disease pathogenesis. These mutations now shape the landscape of targetable treatment options for this patient population. Additionally, previous research has shown that patients with ECD have a higher frequency of concomitant myeloid neoplasms as well as mutations traditionally related to clonal hematopoiesis detected in peripheral monocytes. The goals of this study are to examine treatment outcomes with the use of small molecule inhibitors versus conventional therapy in ECD over a longitudinal period. We also aim to identify relevant trends in laboratory parameters, specifically peripheral blood monocytes, that may offer insight into the mechanisms driving disease progression.

**Abstract:**

A retrospective analysis of 20 adult patients with histopathological and clinical diagnoses of ECD was conducted at a single institution over a twenty-year period (2002–2022). Clinical responses were compared on the basis of treatments rendered, which included chemotherapy, immunotherapy, systemic corticosteroids, surgery and radiation, or targeted agents, referring to any small molecular inhibitors. Treatment response evaluation varied by the anatomic site(s) of disease, the extent of disease at diagnosis, and the imaging modality employed. In this analysis, patients were treated with a combination of targeted agents, myelosuppressive therapies, and radiation at various points in their disease courses. Of these, the most common treatment modality rendered was targeted therapy, employed in 11 of 20 patients. Partial responses or better were observed in 15 of 20 patients. Rates of stable disease trended towards being more frequent with targeted therapy versus conventional therapy but did not reach significance (*p* = 0.2967). Complete response rates trended towards being more common with conventional therapy than molecular (*p* = 0.5) but were equivocal overall. Trends of peripheral blood absolute monocytes with relation to disease activity were reviewed as recent literature implied that monocyte levels surrounding disease progression were of potential prognostic significance in histiocytic diseases. Amongst the patients who progressed at any point during their treatment course, absolute monocyte count (in K/µL) was identified at the closest available timepoint prior to or following disease progression and at the lowest value (nadir) following re-institution of therapy prior to any additional agent(s) being employed. There was no statistically significant difference in either of these monocyte values nor in disease outcomes with respect to treatments rendered within our cohort. However, our cohort consists of a heterogenous population of patients with ECD with data that highlights several trends over a longitudinal period, spanning the advent of targeted therapy. Significant differences are anticipated in ongoing analyses.

## 1. Introduction

Erdheim–Chester disease (ECD) is a rare, non-Langerhans cell histiocytosis classified as a hematopoietic neoplasm in 2016 by the World Health Organization based on the identification of clonal signatures within patients [1,2,3]. ECD was first classified as a “lipoid granulomatosis” in 1930 and later evolved to be recognized as a non-Langerhans cell histiocytosis. For many years, ECD was recognized as an inflammatory or reactive condition. The evolution of this disease from a clinical and scientific standpoint now spans the era of targeted molecular therapy, further shaping its understanding and management. Despite 1500 documented cases in the literature through 2019, there continues to be a scarcity of data on ECD, lending to the heterogeneity of diagnostic evaluation and management [4,5,6]. ECD is characterized by the infiltration of foamy macrophages, surrounding fibrosis or xanthogranulomatosis within tissues and organs throughout the body, including the central nervous system, retroperitoneum, long bones, and large vessels. More life-threatening presentations often have a component of central nervous system or cardiac involvement [1]. Clinical presentation can be widely variable based on anatomic involvement. A diagnosis of ECD is made based on cumulative clinical, radiologic and histologic findings [6,7]. Touton giant cells are occasionally seen on microscopic analyses. The classical immunohistochemistry profile in ECD histiocytes is CD 68+, CD 163+, factor XIIIa+ and CD1a− [6]. Clonal signatures vary and are not required to make a diagnosis but commonly include mutations in the MAP/kinase signaling pathway, including *BRAF*, *MAP2K1*, and *N/KRAS* [1,4,7].

Prior to the advent of molecular therapies, conventional therapy for ECD consisted primarily of chemotherapy or immunosuppressive therapy and was associated with mortality rates up to 60% at three years from diagnosis. Subsequently, interferon-alpha became the standard therapy, resulting in modest improvements in survival outcomes [8]. However, IFN-alpha is often poorly tolerated and ineffective in more severe forms of the disease, including in patients with CNS and cardiac involvement [9]. More recently, in the era of targeted molecular therapy, Ras/MAP kinase pathway inhibitors have demonstrated efficacy in inducing disease remission and improving mortality rates [8,9,10,11,12,13,14,15,16]. Such agents, including the BRAFV600E inhibitor, vemurafenib, and the MEK 1/2 inhibitor, cobimetinib, have gained FDA approval for Erdheim–Chester in 2017 and 2022, respectively. Our study specifically aimed to evaluate treatment outcomes in patients treated with conventional therapy versus targeted molecular therapy over a longitudinal period. Additionally, we aimed to evaluate if peripheral monocytes could be used as a marker for disease activity and response to therapy.

## 2. Materials and Methods

A retrospective analysis of 20 patients diagnosed with ECD at a single institution over a twenty-year period (2002–2022) was conducted. Inclusion criteria included a histopathologic diagnosis of ECD without secondary malignancies within the same microscopic field obscuring the diagnosis. Data pertaining to the supporting histological evidence of ECD for each patient are included in Appendix A. The diagnosis of ECD in this cohort (or in any patient, for that matter), was based on collective characteristic clinical, radiographic, and immunohistochemical findings (for instance CD1a, CD68, and CD163 positivity) and was not solely derived from tissue analysis.

Clinical information collected included biological sex, age at diagnosis, molecular testing (if utilized), immunohistochemistry, imaging results, therapies administered and best treatment outcomes. Clinical, imaging, and molecular data were translated into binary data (“yes” or “no”) for descriptive statistical analysis. Comparison of outcomes with respect to treatment type, time to diagnosis, and use of targeted agents was analyzed. A comprehensive summary with demographic data, diagnostic imaging modalities, molecular testing used (if applicable), treatment(s) rendered, and disease responses is included in Appendix A.

Mean values were generated for continuous variables and categorical variables were summarized as frequencies or proportions using GraphPad Prism 9 © software. Continuous variables included age at diagnosis and time of presentation to diagnosis (in months). Categorical variables included biologic sex, whether molecular diagnostics were utilized (including which techniques and associated results), disease presence by imaging (if so, which modality), anatomic site(s) of involvement, treatment(s) received, and best treatment response (if available). Central nervous system (CNS) involvement was defined by the presence of lesions involving the meninges and brain parenchyma; craniofacial structures (e.g., maxillae, mandible) were not included under this designation.

Treatment response assessments varied for each patient based on several factors, including site of initial disease and imaging used to diagnose and monitor disease activity. To standardize patient outcome assessments in our retrospective analysis, we employed three specific criteria including clinical, PERCIST or RECIST (Table 1), when applicable [17,18]. The response assessment utilized for each of the patients in this cohort is documented in Appendix A. Clinical outcomes were evaluated by the authors in conjunction with the treating hematologist and diagnostic radiologist. All radiological studies referenced herein pertaining to the initial diagnosis and subsequent management of patients with Erdheim–Chester were reviewed during the period of care by a radiologist. Imaging demonstrating comparative pre- and post-treatment responses, where relevant, could not be included for individual patients; however, the Appendix A provides detailed information on each patient’s treatment course.

Treatment response assessments (rates of stable disease, partial response, or complete responses) were then analyzed with respect to treatment administered. Given the nature of this retrospective study, patient records were objectively examined for clinical disease characteristics, diagnostics utilized, treatment(s) administered, and response(s) achieved, as detailed above. Concurrent autoimmune disease was uncommon (only one patient had rheumatoid arthritis and was on methotrexate) and was thus not included as a discrete data point in this analysis. The swimmer’s plot in Appendix A, depicts disease responses with respect to treatments rendered throughout each patient’s clinical course. This information is also included in a narrative format under the “Treatment Summary” column in Appendix A.

Finally, the peripheral blood absolute monocyte count (in K/µL) was documented at multiple points including prior to initiation of therapy, at the time of disease progression, after reintroduction of therapy (when applicable), and at the most recently available (i.e., present) timepoint.

## 3. Results

Of the 20 patients in this analysis, 9 were female and 11 were male (Figure 1A). Age at diagnosis ranged from 14 to 78 years old (mean, 51.2 years). Imaging studies, including CT, MRI, and whole-body PET CT were employed in 7, 6, and 5 patients, respectively. Extranodal primary disease was common (18 patients) specifically with osseous (12), dermal (10), the central nervous system (8), and cardiac (4) structure involvement. Six patients did not have evidence of bone involvement at the time of diagnosis (Figure 1B). Eight patients were diagnosed within 12 months of symptom onset, while 9 patients were diagnosed > 24 months from symptom onset (Figure 1C).

Molecular diagnostics or surrogates for molecular diagnostics in immunohistochemistry were utilized in all 20 patients, (immunohistochemistry for *BRAF* in 18, sequencing for *BRAF* in 4). *BRAFV600E* mutation was detected in eight patients, seven of which by IHC and one by molecular diagnostics. Collectively, the most expressed cell surface markers by immunohistochemistry were in CD68 (8), followed by CD163 (7) and S100+ (5) (Figure 2A).

Regarding disease management, targeted molecular therapy was utilized most often (11 patients), followed by myelosuppression or steroids (8), immunotherapy and radiation (2) (Figure 2B). Second line treatment was required in eleven patients due to progressive disease, with seven of these patients also experiencing side effects from initial therapy. Of the eleven patients with progressive disease, three patients received targeted molecular therapy as first line treatment. Targeted therapy was discontinued in two of these patients due to toxicity and one patient due to progressive disease. Of these eleven patients, five patients had CNS or cardiac involvement at the time of initial diagnosis. Second line therapy selection was widely variable, with targeted molecular treatment being used after retrospective identification of *BRAFV600E* mutation in two patients. Other second line agents included cladribine (*n* = 2), pembrolizumab (1), local radiation therapy (2), binimetinib (1), cobimetinib (1), and dabrafenib (1). One patient with advanced CNS involvement progressed following vinblastine, rituximab, lenalidomide, clofarabine, and local radiation therapy. Mortality rate was 0% in this cohort at the time of treatment analysis.

Amongst the patients with MAP Kinase pathway mutations, one patient was identified to have a t (1;7) *RNF11-BRAF* fusion, somatic diploid *GNAS* amplification, *WT1* and *WNT2* positive, but did not have germline mutations. The other patient was identified to have a Tier II (variant of potential clinical significance) *NRAS* mutation c.182A>G, p.Q61R, at a variant allele frequency of 5% within the bone marrow and spleen. Both patients with MAP kinase mutations were negative for the *BRAFV600E* mutation (Figure 2C).

Regarding outcome assessments, management was designated as “molecular” in patients who received any small molecule inhibitor targeting the Ras/MAP kinase pathway and “conventional” for all other non-targeted therapies, including surgery, radiation, and myelosuppressive therapy (immune/chemotherapy). Responses were then evaluated with respect to therapy rendered. Data presented henceforth will be reported as raw values but then depicted graphically as percentages to allow for direct comparison of responses, since the total number of patients in each category (e.g., partial and complete responders) differed.

Treatment responses were analyzed with respect to type of treatment rendered (conventional versus targeted molecular therapy), as well as time to diagnosis. Best treatment response with respect to active therapy at that time was utilized for the data that follows. This is important to note, as several patients were treated with both conventional and targeted molecular therapy.

Stable disease was observed in five patients out of ten treated with targeted therapy, compared to one patient out of five treated with conventional therapy (*p* = 0.2967). A partial response was observed in one patient out of five treated with conventional therapy and one patient out of ten treated with targeted therapy (*p* = 0.6221). A complete response was observed in three patients out of five treated with conventional therapy and four patients out of ten treated with molecular therapy (*p* = 0.5). Stable disease was observed in two patients diagnosed early (<12 months from symptom onset) and similarly in two patients diagnosed late (>24 months from symptoms onset). Complete response was observed in four out of six patients diagnosed within 12 months of symptom onset, compared to three out of six patients diagnosed >24 months after symptom onset. In total, there were three patients with unknown disease response (Figure 3). Due to the small sample size, the raw values presented are not suggestive of significant differences within this cohort. The timeframe of response assessments for each patient are included in Appendix A.

Finally, an evaluation of peripheral blood absolute monocytes in all patients who had progressed in therapy identified a decrease in pre-progression absolute monocytes compared to post-progression monocytes after re-initiation of therapy by a mean value of 0.2 K/uL, which did not reach significance (*p* = 0.3125, 95% CI −0.2373 to 0.6373, Figure 4A). A trend toward decrease in pre-therapy absolute monocytes compared to post-therapy monocytes was observed as well, but a significant difference was not reached (averages of 0.49 and 0.33, respectively; *p* = 0.0988, 95% CI −0.0302 to 0.3408, Figure 4B).

Regarding outcomes beyond treatment responses, data on therapy-related toxicities, duration of treatment and reasons for discontinuation (where applicable) are summarized for each patient in Appendix A**.** Long-term sequelae stemming from cytotoxic and targeted therapies, to date, have not been observed in this cohort, nor have secondary lymphoid myeloid neoplasms. The mean time from date of diagnosis to most recent follow-up was 109.9 months.

## 4. Discussion

Our current study highlights several challenges in diagnosing, treating, and reporting disease outcomes in a rare and heterogenous disease. Despite this, we found trends in the data to suggest that even in patients treated with several different therapies, higher rates of stable disease were observed more frequently with targeted therapy compared to conventional. While we observed higher complete response rates in patients treated with conventional therapy, we attribute this to the relatively short follow-up periods since initiation of targeted therapy compared to that of conventional therapy. We acknowledge that given the small cohort size, statistical significance was not observed with these outcomes; however, we find value in reporting trends in disease outcomes given the rarity of the disease. Furthermore, the relatively new recommendation for use of targeted therapy as first line therapy in ECD necessitates longitudinal analyses like these, to better understand disease monitoring, outcomes and side effects of therapy.

The current proposed diagnostic and treatment pathways for patients newly diagnosed with ECD, per the consensus guidelines published in 2019, recommends BRAF testing as the first branchpoint in diagnosis [2,8,10]. Supporting data for these guidelines included several studies demonstrating recurring mutations in BRAF V600E and other activating alterations in the MAP kinase and PI3K-AKT pathways [1,2,8,9,10,11,12,13,14,15,16]. Based on these guidelines, it is recommended that if a patient has mild *BRAF V600E*-positive disease, then targeted BRAF inhibition or conventional therapy (INF-alpha versus other myelosuppressive agent) should be implemented as first line treatment. Cohen Aubart et al. recommends use of targeted molecular therapy with vemurafenib or dabrafenib first line [1,8,10] in patients with moderate/severe ECD (i.e., cardiac or neurologic involvement) or with end organ involvement. In patients with *BRAF* WT and moderate/severe disease, they recommend evaluation for other *MAPK-ERK* mutations. Notably, treating empirically with MEK inhibitors in patients with acutely severe ECD without an identifiable *MAPK* mutation is also a reasonable strategy [19,20,21,22,23].

Since these recommendations have been published, few studies have been conducted evaluating treatment efficacy, outcomes, and tolerability of BRAF and MEK inhibitors in ECD [5,8,9,10,11,12,13,14,15,16]. A study conducted in 2017 by Cohen Aubart et al., was the first to demonstrate vemurafenib efficacy with long-term follow-up in a cohort of 54 patients who were included in the French Histiocytosis Registry. Findings here included an 88% response rate of partial metabolic response (PMR, n = 35) or complete metabolic response (CMR, n = 7) at six months after treatment initiation [1,8,15]. Collectively, this study was able to follow a cohort of patients treated solely with targeted molecular therapy, whereas ours followed a more diverse patient population with respect to treatments rendered. Furthermore, we evaluated disease response with both clinical exam and imaging, rather than imaging alone. This is important to note, as reduced FDG-avidity on PET-CT does not always correspond to complete disease regression due to the extensive fibrosis often seen in this disease and the persistent end-organ histiocytic infiltration despite metabolic responses on imaging [8]. Most importantly, we were able to compare treatment outcomes of molecular versus conventional therapies, although we acknowledge that statistical significance was not observed due to the small sample size within our cohort.

Another observational cohort study of 60 patients with ECD enrolled in the National Human Genome Research Institute clinical protocol from 2011 to 2015 reported on clinical outcomes with both conventional and targeted molecular therapy in ECD [7]. The majority of their patients were treated with conventional rather than targeted therapy (23 versus 8 patients, respectively). Here, clinical examination and radiographic imaging were both utilized to define therapeutic response, making this an ideal comparator to our cohort. Their findings differ from ours, in that they observed higher rates of stable disease in patients treated with conventional therapy versus targeted therapy, a difference likely explained by a smaller number of patients treated with targeted therapy within their cohort. Both studies also demonstrated higher relapse rates with conventional therapy when compared to targeted agents. Our study also differs in that we were able to follow our patients over a twenty-year period compared to five years.

In further exploring the landscape of targeted therapy evaluation in ECD, several trials have been designed to better understand the efficacy of BRAF inhibitors. A systemic review published by Aziz et al. in 2022, for instance, identified three clinical trials each aiming to identify treatment efficacy of vemurafenib in patients with ECD [5]. This review included data from three sources: an open-label non-randomized phase two clinical trial of 22 patients from Diamond et al. (2018), a cohort study of 122 patients (18 of whom had ECD or LCH) with the *BRAFV600E* mutation (including patients with non-melanomatous cancers) being treated with vemurafenib by Hyman et al. (2015), and finally, an open study of 8 patients with ECD with either CNS or cardiac involvement treated with vemurafenib by Haroche et al. (2015) [21]. These three trials each demonstrated treatment response to vemurafenib to varying degrees [5,21,22,23].

Critical evaluation within the ECD literature has also brought to our attention a lack of standardized use of sensitive genomic sequencing techniques in the diagnostic evaluation of patients with ECD. One recent systematic review across 760 studies, 133 articles, and 311 ECD patients emphasized the importance of implementing such evaluations when approaching the disease [19]. Authors reported, for instance, on *BRAF* mutations as being associated with neurologic disease (183 of 273 patients, 67%, *p* < 0.001), *KRAS* and *NRAS* with cutaneous and pleural involvement (583 and 44% of patients, respectively), and *MAP2K1* with peritoneal and retroperitoneal lesions (4 of 11 patients, 36.4%, *p* = 0.01); *PIK3CA* was not associated with specific organ involvement. This study uniquely stressed the influence of a possible baseline early mutation within monocytes within ECD and further argues for a thorough genetic analysis in these patients [11,13,19]. In an effort to bring these findings into clinical practice, the 2020 consensus ECD guidelines recommended the use of sensitive genetic sequencing techniques in standard polymerase chain reaction, droplet digital PCR or targeted-capture next-generation sequencing in samples with negative or equivocal immunohistochemistry (IHC) for *BRAFV600E*, arguing that IHC is not sensitive for evaluation of mutational status in ECD tissue. Authors also recommended that NGS be used in patients without the *BRAFV600E* mutation to identify additional targetable mutations within the MAP-ERK and PI3K-AKT pathways (*KRAS*, *NRAS*, *ARAF*, *RAF1*, *MAP2K1*, *MAP2K2*, *BRAF* indels, and *PI3KA*) [5,10,19].

Lastly, there are multiple potential etiologies to explain the relative (albeit insignificant to-date) peripheral monocytosis at the time of disease progression, followed by a relative decrease at time of re-institution of therapy observed within our cohort. One explanation is that, in response to a progression in tumor burden, monocytes expand and differentiate into dendritic cells, which upregulate major histocompatibility class II expression [24]. Another possibility is that peripheral blood monocytes themselves are simply a component of circulating disease and expand as ECD is progressing. Prior examinations into histiocyte ontogeny would support this theory. Specifically, research has suggested that histiocytic disorders arise from pre-existing clonal hematopoiesis whereby clonal hematopoietic progenitors harboring *TET2* mutations later differentiate into circulating monocytes that acquire *BRAF* mutations and circulate as ECD [25,26,27,28,29]. With this theory, relative peripheral monocytosis at time of progression, followed by reduction while on therapy, might be reflective of patients’ ECD responding to therapy.

The authors recognize that the monocyte data provided herein is not significant, owing to small study numbers. We wish to acknowledge even these negative findings as the trends are relevant and are expected to reach significance in ongoing analyses within our cohort. This is further supported by similar findings seen within histiocytic diseases in 2023 studies by Razanmahery et al. and Reynolds et al. [28,29]. While these trends remain under expansive investigation, the aforementioned studies and cumulative clinical data now serve as the basis for ECD management as set forth by the guidelines from the National Comprehensive Cancer Network [30].

## 5. Conclusions

To our knowledge, this study is one of the first of its kind to compare treatment outcomes of both conventional and targeted therapy in adults with Erdheim–Chester disease. Based on our data, early recognition of targetable mutations and utilization of targeted therapy are both associated with higher rates of disease stability and at least a partial response to therapy by clinical exam and imaging. Within our cohort, stable disease was more often observed in patients treated with targeted therapy whereas complete response was observed more often in patients treated with conventional therapy. This is likely due to relatively short follow-up periods since initiation of targeted therapy in this cohort, compared to that of conventional therapy. We expect higher complete response rates with targeted therapy in ongoing analyses with longer periods of follow-up. It is also important to note that this study spans a large period of 20 years, during which time the landscape of treatment for ECD has changed significantly, spanning the introduction of molecular diagnostics and targeted molecular therapies. With that, we acknowledge the potential for confounding within our data set. Separately, in a correlative translational study, we plan to perform next generation sequencing of our existing cohort of patients with ECD and identify patterns between the extent of mutational heterogeneity (as well as the exact mutations) in ECD and how they relate to clinical outcomes. Finally, the findings on relative monocytosis are intriguing and require cohort expansion to determine clinical significance with regard to its use as a prognostic tool.

## Figures and Tables

**Figure 1 cancers-16-01299-f001:**
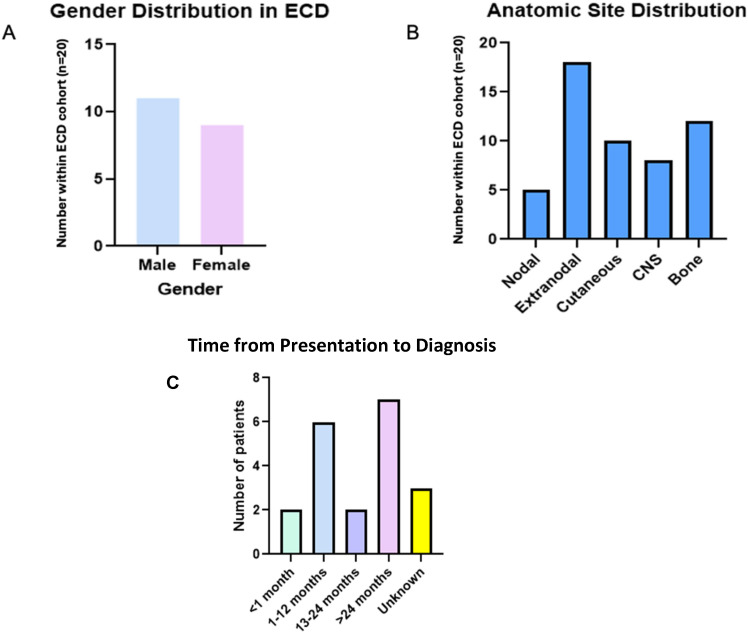
(**A**). Males diagnosed more frequently than females (11 versus 9 patients, respectively), which is consistent with known disease phenotype. (**B**). Disease site varied, with extranodal lesions being most common. (**C**). Time from initial presentation to diagnosis of ECD.

**Figure 2 cancers-16-01299-f002:**
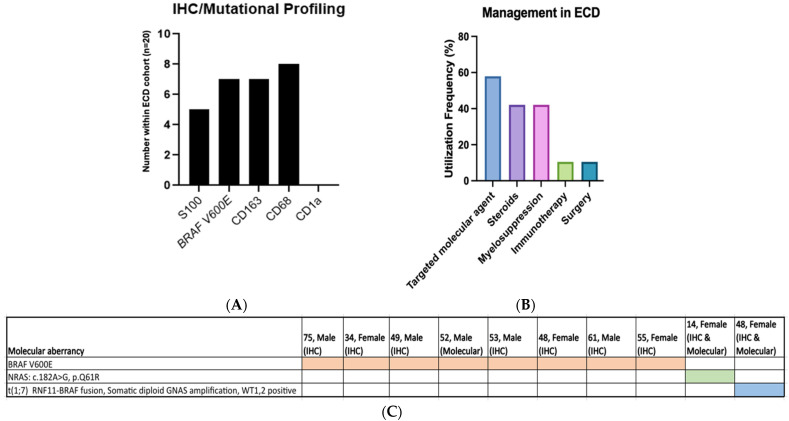
(**A**). Cumulative immunohistochemistry staining highlights propensity for BRAF V600E mutation and CD68 cell surface marker within ECD. (**B**). Targeted molecular therapy was utilized most often within our cohort of patients. (**C**). Depiction of mutations observed within our cohort of ECD patients. MAP kinase pathway related mutations, specifically *BRAFV600E*, were most common. Sequencing data is limited within our cohort to four patients total, three of which had mutations identified, as depicted above.

**Figure 3 cancers-16-01299-f003:**
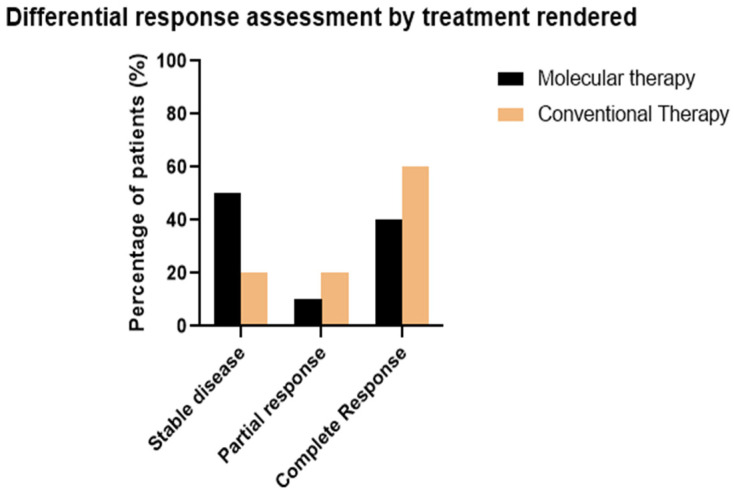
Differential response assessment by therapy administered.

**Figure 4 cancers-16-01299-f004:**
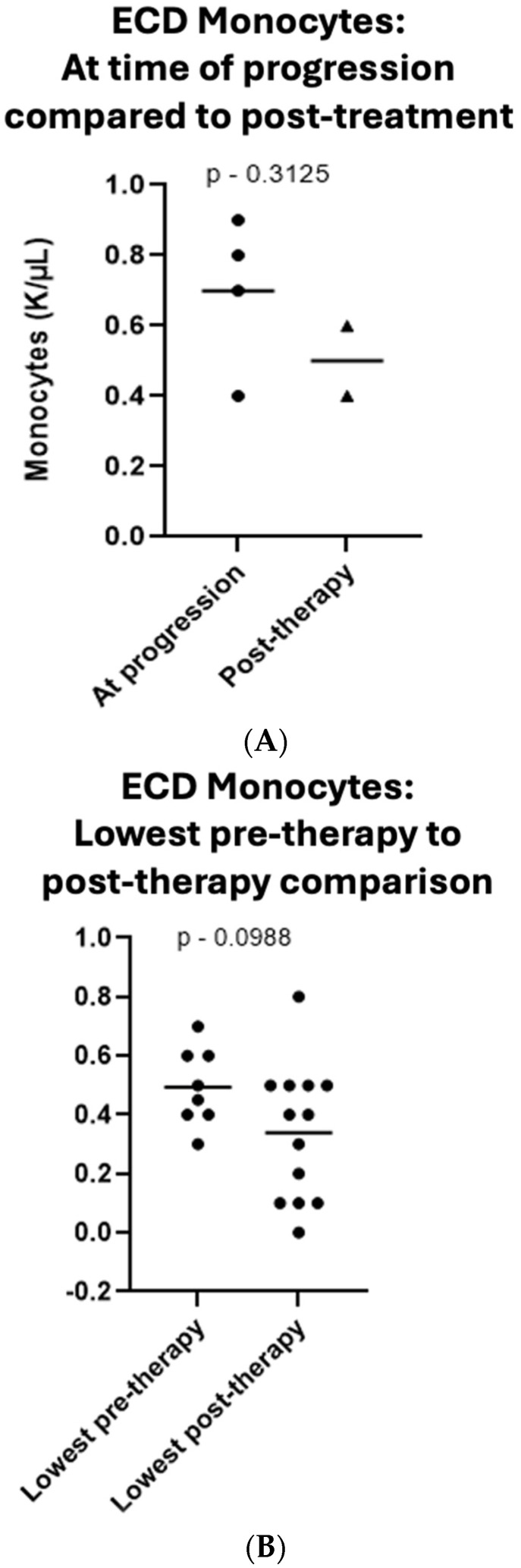
Peripheral monocytes decreased by an average of 0.2 K/µL between disease progression and following re-initiation of therapy (**A**). There was an average decrease by 0.16 K/µL, between pre-therapy absolute monocytes compared to post-therapy monocytes (**B**).

**Table 1 cancers-16-01299-t001:** Response assessment criteria for this retrospective study ECD = Erdheim–Chester disease. Physicians performed objective clinical response assessments. Metabolic responses required comparison between nuclear positron emission tomography (PET)-generated images, with the use of positron emission tomography response criteria in solid tumors (PERCIST). Serial comparison was conducted between either computed tomography (CT) or magnetic resonance imaging (MRI) using RECIST 1.1 criteria (response evaluation criteria in solid tumors) when applicable. SUL = standardized uptake by lean body mass, TLG = total lesion glycolysis.

Response Assessment Tool	Most Utilized in	Stable Disease (SD)	Partial Response (PR)	Complete Response (CR)	Progressive Disease (PD)
Clinical	Localized ECD	Grossly unchanged by serial physical examination and/or serial photoimaging or laboratory values	Grossly regressed by serial physical examination and/or serial photoimaging	Grossly resolved by serial physical examination and/or serial photoimaging	Grossly enlarged from prior baseline by examination and/or serial photoimaging
PERCIST [17]	Disseminated disease by PET	Not meeting other criteria	>30% decrease (minimum 0.8 units) in SUL peak	Normalization of all lesions to an SUL equal to surrounding tissue and mean in liver	>30% increase (minimum 0.8 units) in SUL peak or >75% TLG increase in the 5 highest metabolically active lesions
RECIST 1.1 [18]	Disseminated, visceral disease or neurologic disease by CT and/or MRI	Not meeting other criteria	Sum ≥30% decrease in diameter of target lesion	Complete disappearance of any target lesions	Sum ≥20% increase in diameter of target lesion

## Data Availability

Primary data included patient-specific details including treatments and outcomes. These are provided in Appendix A. These data were used to generate supporting data within this manuscript.

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
