# Peer review of "Erdheim–Chester Disease: Investigating the Correlation between Targeted Treatment Therapy and Disease Outcomes"

_cancers, 2024, doi:10.3390/cancers16071299_

Round 1
Reviewer 1 Report
Comments and Suggestions for Authors
This is a very well written single institutional series of Erdheim-Chester disease (ECD) describing the experience related to disease presentation and involvement as well as treatment.
One minor comment I have to improve the manuscript is to describe the outcomes beyond treatment responses, for example, toxicities, treatment discontinuation, development of second cancers, or infectious complications. May also include the median and range of follow-up. There are some descriptions of these in Figure 6 but not in the narrative under Results. A brief description would be relevant.
Author Response
Thank you very much for taking the time to review our manuscript. Your thoughtful feedback was greatly appreciated. Please see the below updates to our manuscript based on the feedback provided:
One minor comment I have to improve the manuscript is to describe the outcomes beyond treatment responses, for example, toxicities, treatment discontinuation, development of second cancers, or infectious complications. May also include the median and range of follow-up. There are some descriptions of these in Figure 6 but not in the narrative under Results. A brief description would be relevant.
Response: At the end of the results section, there is now reference to this requested information, specifically as to adverse effects/toxicities or progression being the most common reasons for treatment discontinuation. We have also now noted in the text that secondary malignancies stemming from the receipt of targeted or cytotoxic therapies, including myeloid neoplasms, were not observed in this cohort.
Reviewer 2 Report
Comments and Suggestions for Authors
Thank you for the opportunity to review this manuscript by Wilcox and colleagues. The authors report the clinical outcomes of 20 patients with ECD seen and treated over a 20-year period.
Abstract:
“In this analysis, 11 of 20 of patients were treated with targeted therapy, followed by myelosuppressive therapy and steroids, immunotherapy and then radiation” – were all 11 patients treated with these modalities in this order?
“In all patients who progressed on therapy, we identified a lower absolute monocyte count between the time of disease progression when compared to re-initiation of therapy.” - clarify what two times you are referring to with “between” here.
There are limited numbers included in the abstract, and no p values– please include some numbers to give magnitude of the effects or observations mentioned.
Introduction:
Recommend mentioning cobimetinib as being approved for treatment.
Materials and methods:
Who reviewed the radiology? Was a radiologist involved?
How were clinical outcomes reviewed? Consensus, or by one author?
How was CNS disease defined? Did it include calvarial lesions, dural based lesions, meningeal disease, or only parenchymal disease?
In case 4 – cervical/spinal involvement is noted – is this vertebral involvement or spinal cord?
Results:
“The most common mutations identified were BRAFV600E (7) and CD68 (8), followed by CD163 (7), S100+ (5) and MAP Kinase pathway mutations (2) (Figure 2A).” - I presume the authors are referring to CD68 positive cells on pathology, rather than mutations in the CD68 gene? Similar for Figure 2A.
“Mortality rate was 0% in this cohort at the time of treatment analysis.” – please include follow up time here.
The timeframe for response needs to be included with all analyses. Statements of superiority of one treatment over another need to be accompanied with a supporting statistical test. E.g. 3/10 vs 4/10 does not imply a significant difference.
“There was no difference in rates of stable disease in patients diagnosed early (< 12 months from symptom onset), versus late (> 24 months from symptoms onset).” – please include absolute numbers.
“an evaluation of peripheral blood absolute monocytes in all patients who had progressed on therapy identified a decrease in pre-progression absolute monocytes compared to post-progression monocytes after re-initiation of therapy by a mean value of 0.2 182K/µL, although results were imprecise (p = 0.3125, 95% CI -0.2373 to 0.6373, Figure 5A)” – this analysis is based on data from 2 patients in the post therapy group, and appears to be an unpaired analysis, rather than a paired analysis, which would be required. It is also not statistically significant. Similarly, for the second analysis – the data shown do not support a definite drop in monocytes.
Discussion:
Recommend including the NCCN guidelines in discussion.
Figures:
Figure 1 A and B x axis is labeled as “number within RDD cohort (N=35)”. – is this correct?
Figure 1A legend “males more commonly diagnosed, consistent with known phenotype” - the difference does not look to be statistically significant from the graph. Was a statistical test applied?
Figure 1D includes percentages, but not absolute numbers. This would be better visualized as a bar chart with absolute numbers.
Figure 3 - apparently includes both IHC and molecular genetic testing. Please clarify in the figure which patients had which testing modality.
Figure 4 A – please use absolute numbers rather than percentages.
Reviewer 3 Report
Comments and Suggestions for Authors
The manuscript, entitled “Erdheim Chester Disease: Investigating the Correlation Between Targeted Treatment Therapy and Disease Outcomes”, is a very small retrospective study on patients with ECD and aims to compare the impact of small molecule inhibitors with conventional therapy and to identify relevant trends in laboratory parameters that may offer insight into the mechanisms driving disease progression. It is clear the authors have worked very hard on this paper and the supplementary table is very nice. While this study was an interesting read, there are unfortunately many issues that prevent the paper from being published in the current form.
Major
In the “Summary”, the authors report that the goal of this study is to compare the impact of small molecule inhibitors with conventional therapy. However, there are no comparative analyses done. There is no mention of overall survival or progression free survival.
The authors also report that the goal of the study is to identify relevant trends in laboratory parameters, yet only the peripheral monocyte count is investigated and this is not significant. The results were not statistically significant and with such a small sample size, nothing can be concluded from this.
In Figure 1A and Figure 1B, the y-axis of the figures state “number with RDD cohort (n=35).” This is a serious error and the authors need to double check that they are using the right cohort for the analysis of the current study.
The symbols on the swimmers plot is very difficult to read. Please make the figure symbols larger or use a higher quality image.
Minor
References are not in chronologic order. For example reference 5 does not get mentioned until later in the paper and other references such as 7 and 8 come before reference 5
Is the reference for line 61-63 correct? The paper that is being referenced discusses sirolimus and line 61-63 is talking about interferon-alpha.
Is the reference for line 64-66 correct? The paper that is being referenced discusses sirolimus and line 64066 is talking about BRAF inhibitors.
Please arrange for Figures to be in chronologic order as they appear in the text. Also, on line 117, the author’s reference Figure 1B, but I believe this should be Figure 1A. Please double check that Figures are referenced correctly.
I personally prefer using “median” instead of the “mean” as the median is less affected by outliers.
On line 121, the authors mention that 7, 6, and 5 patients respectively had CT, MRI, whole-body PET-CT. Then in Figure 1D, they mention 55.5% had PET-CT, 25% had MRI, and 13.89% had CT. The %s in Figure 1D and the numbers in the text do not match.
On line 131, the authors state that “molecular diagnostics were utilized in all 20 patients (immunohistochemistry in 18, sequencing in 4).” It should be noted that immunohistochemistry is not a molecular diagnostic. Immunohistochemistry is used as a surrogate for molecular diagnostics.
On line 132, please specify how many of the “BRAFV600E” was detected by IHC vs. NGS
On line 132, it should be noted that CD68, CD163, S100+, are not mutations but IHC markers.
Overall Impression:
There have been multiple retrospective studies conducted at this point in time on ECD. The authors have a very small sample size (n=20) and do not perform any comparative analyses. While it is interesting that the authors want to investigate the significance of the absolute monocyte count, the small sample size precludes the ability to draw meaningful conclusions and resulted in a non-significant p-value. As things currently stand, the study unfortunately does not provide additional value to the existing literature. There are also multiple serious errors in the manuscript as mentioned above. At this point in time, I cannot recommend publication of this manuscript. If all of the issues are addressed, then perhaps the manuscript can be reconsidered for publication as a case series.
Reviewer 4 Report
Comments and Suggestions for Authors
The authors conducted an interesting single institution outcomes study that investigated the correlation between targeted treatment therapy and disease outcomes in Erdheim-Chester disease (ECD). The retrospective methods were reasonable for this outcomes study. The figures were somewhat informative and supported the discussion and conclusions. This outcomes study on targeted therapy in ECD will be of interest to physicians and scientists who study histiocytic neoplasms and manage histiocytosis patients. I have some minor comments to help improve the clarity of the data presentation and discussion of this study.
1. The bar chart in Figure 2A would be better if the immunohistochemistry results and mutational results were separated into different charts. Also, Figure 3 would be better presented as a subpanel of Figure 2 to further explain the mutational bar chart. Furthermore, the figure legend wording in Figure 2A and the results discussion around Figure 2 are incorrect and somewhat confusing. BRAF V600E and other MAP kinase pathway mutations is correct; however, CD68, CD163, and S100 are immunohistochemical stains and surface markers and not mutations as they seem to be called in the text.
2. Minor comment: Table 1 needs to be revised to where the column headings do not use hyphenated words. If a word cannot fit on the first line, then it should be moved to the second line. This will improve the readability of the Table. This also goes for words in the rows beneath the column headings.
3. Minor comment: I would also recommend placing the panel letters in Figure 1 above the upper left corners of the individual subpanels because the current lettering can be distracting and a bit hard to follow for the readers.
4. Minor comment: Please carefully proofread the manuscript because there are several grammatical, punctuation, and spelling errors scattered throughout the manuscript. There are some run-on sentences, subject-verb agreement issues, and typographical errors involving gene symbols that can be distracting to readers. Also, all human gene symbols referring to DNA mutations. Also, "PI3K-AKT" pathway is more commonly used and accepted in formal scientific writing than "PI3-AKT pathway.
Comments on the Quality of English LanguageMinor comment: Please carefully proofread the manuscript because there are several grammatical, punctuation, and spelling errors scattered throughout the manuscript. There are some run-on sentences, subject-verb agreement issues, and typographical errors involving gene symbols that can be distracting to readers. Also, all human gene symbols referring to DNA mutations. Also, "PI3K-AKT" pathway is more commonly used and accepted in formal scientific writing than "PI3-AKT pathway.
Reviewer 5 Report
Comments and Suggestions for Authors
Wilcox et al present the outcomes of 20 Erdheim Chester Disease (ECD) patients seen over a 20 year period at a single institution and treated with a wide range of therapies. The manuscript highlights the inherent difficulty in summarizing heterogeneous treatment results in this population. Page 3, line 124 it is stated that 6 patients did not have bone involvement (Figure 1D, should be 1B). On page 4 the authors describe molecular diagnostic characteristics, CD68, CD163, and S100 are not mutations, but cell surface markers. Patients were treated with MAPK inhibitors, classical chemotherapy, immunotherapy, surgery and radiation therapy. It is not clear if the results of each therapy was used to tally the totals, or if only the last treatment type was used for this leading to Figure 4A. On page 5, line 152, 3 patients on targeted therapy had progressive disease. Was this because the targeted therapy was stopped? Figures 1C and 4B are redundant.
Figures 5A and B show no significant differences. The monocyte topic should be eliminated from the paper
Figure 6 is impossible to read. The therapy symbols can not be distinguished from the background colors. Need large white symbols for types of therapies.
The discussion is way too long. Do not summarize results from other papers.
Supplemental Table 1. Descriptions of molecular testing, histology and treatment summaries are excessively long and often not helpful. Response assessment should include change in SUV or SUL. Ultimately 11/20 had unknown responses because of being lost to follow up. This makes it impossible to understand the real outcomes. Patient 5 had PERCIST evaluation listed, but no PET scan was done according to the imaging modality notation. The authors should be more clear that a biopsy does not make the diagnosis of ECD, but rather the clinical findings are supported by the xanthomatous or fibrotic histology of cells staining with CD68 or CD163 and not CD1a.
The swimmer plots do not correlate with the written descriptions in table S1. Maybe this is because the symbols for treatments are so hard to read. But also the length of the swimmer plot bar and the written words do not seem to be consistent.
Patient 15. If this patient is really a CR you should show the CT images. It is hard to believe a patient with retroperitoneal and pericardial fibrosis really was a CR or maintained it 15 months off therapy.
Page 5, lines 171: How can you state that higher partial response rates were observed more observed more frequently in conventional therapy than targeted when 1 patient was listed for each?
Likewise lines 175 to 177 describes CR rates in those with early vs later diagnosis. The difference between 4/6 vs 3/6 is meaningless. What were the types of therapies?
Round 2
Reviewer 2 Report
Comments and Suggestions for Authors
Thank you for the opportunity to review this manuscript again.
Abstract:
"Rates of stable disease trended towards being more frequent with targeted therapy versus conventional therapy but did not reach significance (p=0.2967)" - Recommend removing all reference to trends, only highlighting differences which are statistically significant. Additionally, include the absolute value of differences addition to P values, wherever P values are reported.
Trends of peripheral blood absolute monocytes with relation to disease activity 35 were reviewed, based on recent literature implying monocytes having potential prognostic significance in histiocytic diseases. Amongst the patients who progressed at any point during their treatment course, absolute monocyte count (in K/µL) was identified at the closest-available timepoint 38 prior to or following disease progression and at the lowest value (nadir) following re-institution of therapy prior to any additional agent(s) being employed - If this is being included in the abstract, the results should also be included.
Abstract should contain a conclusion.
Results:
Mortality remains reported at 0%, but without a timeframe.
"Stable disease was observed in five patients treated with targeted therapy, compared 212 to one patient treated with conventional therapy (p=0.2967)." - Both the numerator and the denominator needs to be reported for all such statements.
Reviewer 3 Report
Comments and Suggestions for Authors
Revisions are adequate.
Author Response
Thank you for your time in reviewing this manuscript. The feedback was greatly appreciated and added to the quality of the paper.
Reviewer 5 Report
Comments and Suggestions for Authors
Upon further review of this paper it is clear that 5 patients should not be included in the survey. Patients 1, 8, and 10 had unknown outcome according to Supplemental Table A1. Patient 17 and 18 were lost to follow-up. By my count 9 patients were treated with inhibitors, 2 with chemotherapy (leaving out one who had PD an unknown response, 2 with PEG interferon, one with RT and 1 with bisphosphonates. Thus PRs or CRs = 8/15. No, there was no trend of conventional therapy having more CRs than inhibitors. In the last sentence of the abstract you state discuss monocyte counts but don't give any results. The monocyte data is completely useless since the findings are not significant. Such data should NOT be in the literature.
Round 3
Reviewer 5 Report
Comments and Suggestions for Authors
The authors have adequately responded to my comments.